# Exciton-driven antiferromagnetic metal in a correlated van der Waals insulator

Carina A. Belvin [1,6], Edoardo Baldini [1,6], Ilkem Ozge Ozel[1], Dan Mao[1], Hoi Chun Po[1], Clifford J. Allington [1], Suhan Son[2,3], Beom Hyun Kim[4], Jonghyeon Kim [5], Inho Hwang[2,3], Jae Hoon Kim [5], Je-Geun Park [2,3✉], T. Senthil[1] & Nuh Gedik [1✉]

Collective excitations of bound electron-hole pairs—known as excitons—are ubiquitous in condensed matter, emerging in systems as diverse as band semiconductors, molecular crystals, and proteins. Recently, their existence in strongly correlated electron materials has attracted increasing interest due to the excitons' unique coupling to spin and orbital degrees of freedom. The non-equilibrium driving of such dressed quasiparticles offers a promising platform for realizing unconventional many-body phenomena and phases beyond thermodynamic equilibrium. Here, we achieve this in the van der Waals correlated insulator $NiPS_3$ by photoexciting its newly discovered spin–orbit-entangled excitons that arise from Zhang-Rice states. By monitoring the time evolution of the terahertz conductivity, we observe the coexistence of itinerant carriers produced by exciton dissociation and a long-wavelength antiferromagnetic magnon that coherently precesses in time. These results demonstrate the emergence of a transient metallic state that preserves long-range antiferromagnetism, a phase that cannot be reached by simply tuning the temperature. More broadly, our findings open an avenue toward the exciton-mediated optical manipulation of magnetism.

[1] Department of Physics, Massachusetts Institute of Technology, Cambridge, MA, USA. [2] Center for Correlated Electron Systems, Institute for Basic Science, Seoul, Korea. [3] Center for Quantum Materials, Department of Physics and Astronomy, Seoul National University, Seoul, Korea. [4] Korea Institute for Advanced Study, Seoul, Korea. [5] Department of Physics, Yonsei University, Seoul, Korea. [6] These authors contributed equally: Carina A. Belvin, Edoardo Baldini. ✉email: jgpark10@snu.ac.kr; gedik@mit.edu

The prototypical correlated system, known as a Mott insulator, possesses a ground state in which the electrons are localized, one per site, due to their strong Coulomb interaction[1]. Consequently, the electron spins are also localized and must order in some way at low temperature due to entropy considerations. The spins prefer to align antiferromagnetically to minimize the kinetic energy. Starting from this insulating antiferromagnetic ground state, the only phases that can be reached by tuning the temperature are a paramagnetic insulator or a paramagnetic metal. Owing to their inherently intertwined degrees of freedom, correlated materials can display excitons that, unlike their analog in band insulators, couple to spin and orbital degrees of freedom[2,3] or form more exotic composite quasiparticles[4]. Therefore, driving these excitons with light offers a playground for exploring many-body phenomena beyond the physics of band insulators, especially the promise of reaching new phases of matter.

One example that holds intriguing possibilities for exploring the effects of a non-equilibrium population of excitons in a correlated insulator is the recently discovered spin–orbit-entangled excitons in the van der Waals antiferromagnet NiPS$_3$[5]. In this compound, the Ni atoms are arranged in a two-dimensional honeycomb lattice and their magnetic moments exhibit a zigzag antiferromagnetic order below the Néel temperature $T_N = 157$ K[6] (Fig. 1a). Optically, NiPS$_3$ has a charge-transfer gap of ~1.8 eV[7] at low temperature. Below this charge excitation lies a rich spectrum of sub-gap absorption resonances (Fig. 1b), including on-site $d$-$d$ transitions around 1.1 and 1.7 eV and the complex of spin–orbit-entangled excitons around 1.5 eV[5]. These excitons are based on

Zhang-Rice states[8], which consist of a hole spin in a localized Ni $3d$ orbital and a hole spin shared by the $3p$ orbitals of its surrounding S ligands. Through an extensive characterization of their equilibrium properties using multiple spectroscopic probes, these peculiar excitons were assigned to the transition from a Zhang-Rice triplet to singlet and were found to exhibit intrinsic coupling to the long-range antiferromagnetic order[5]. This inherent connection between the excitons and antiferromagnetism presents fascinating prospects for manipulating both the spins and charge carriers of the system upon non-equilibrium driving, an avenue that has not been explored to date.

Here, we photoexcite NiPS$_3$ close to its spin–orbit-entangled exciton transitions well below the charge-transfer gap. By monitoring the low-energy electrodynamics of the system in the terahertz (THz) range, we reveal the presence of an itinerant conductivity due to the dissociation of excitons into mobile carriers, accompanied by the excitation of a long-wavelength antiferromagnetic spin precession. These observations indicate a transient conducting antiferromagnetic state that cannot be achieved by tuning the temperature of the system.

## Results

**Terahertz probing of electronic and magnetic dynamics.** To probe the antiferromagnetic order in NiPS$_3$, we focus on a long-wavelength magnon at the Brillouin zone center[9] whose real-space spin precession is depicted in Supplementary Fig. 16. First, we reveal the equilibrium lineshape of this magnon mode in the THz absorption ($\alpha$) (Fig. 1c). At 5 K, the magnon has an extremely sharp linewidth of ~0.1 meV, which indicates a long

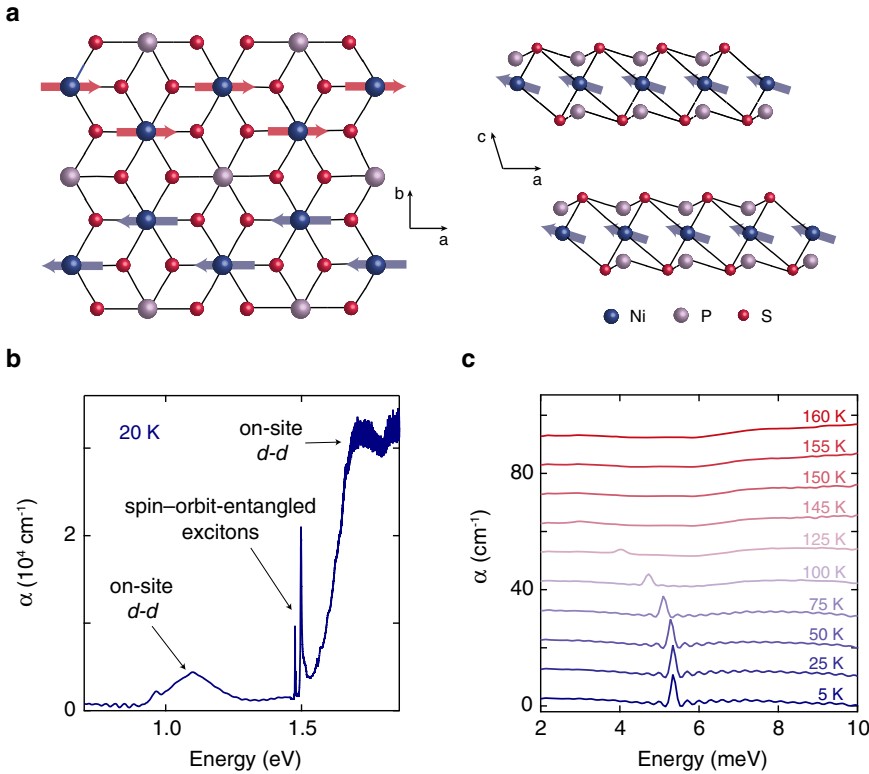

**Fig. 1 Crystal structure and optical properties of NiPS$_3$. a** Crystal and magnetic structure of NiPS$_3$. The Ni atoms (blue spheres) are arranged in a two-dimensional honeycomb lattice in the $ab$-plane. The magnetic moments (red and blue arrows), which point mostly along the $a$-direction, form a zigzag antiferromagnetic pattern with weak ferromagnetic coupling in the $c$-direction. **b** Optical absorption ($\alpha$) of NiPS$_3$ below the charge-transfer gap at 20 K. The features around 1.5 eV are the spin–orbit-entangled exciton transitions reported in ref. [5]. The broad structures around 1.1 and 1.7 eV are on-site $d$-$d$ transitions. The data is adapted from ref. [5]. **c** Absorption spectrum in the THz range at various temperatures. The traces are offset vertically by 10 cm$^{-1}$ for clarity. The low-temperature curves show an absorption peak corresponding to a magnon resonance. The magnon energy softens with increasing temperature and the mode disappears above $T_N = 157$ K.

damping time. Above $T_N$, the mode is no longer present. Its low-temperature energy of 5.3 meV softens with increasing temperature following an order-parameter-like dependence (see below).

Next, we photoexcite the system using a near-infrared laser pulse whose photon energy lies in the spectral region of the spin–orbit-entangled excitons of NiPS$_3$. We simultaneously track the electronic and magnetic degrees of freedom by mapping the change in the low-energy optical conductivity with a weak, delayed THz probe pulse (Fig. 2a). Figure 2b shows the spectro-temporal evolution of the pump-induced change in the real part of the optical conductivity ($\Delta\sigma_1$) at low temperature. The spectral response contains two features: a Drude-like behavior, indicating the presence of itinerant electronic carriers, and the first derivative of a Lorentzian lineshape around the magnon energy. The positive lobe of the narrow derivative-like shape is at lower energies, which signifies that the magnon energy is slightly redshifted compared to its equilibrium value (by only ~2%). No sizeable broadening of the lineshape occurs, meaning that the long-range antiferromagnetic order is preserved. In Fig. 2c, we compare the temporal evolution of the Drude conductivity and the magnon by integrating the response over their characteristic energy regions (1.6–1.8 meV and 5.2–5.4 meV, respectively). We find that the Drude contribution persists for several picoseconds and the magnon exhibits coherent oscillations at the redshifted energy (due to THz emission, which we discuss later). Notably,

these oscillations begin during the rise of the Drude response rather than at a later time after thermalization with the lattice, indicating that a non-thermal mechanism is responsible for launching the magnon coherently. In addition, we infer the nature of the carriers participating in the itinerant response by extracting the Drude scattering rate ($\gamma$) and plasma frequency ($\omega_p$) from the conductivity spectrum (see Supplementary Note 4 for additional details). At the peak of the Drude signal (Supplementary Fig. 7), we obtain $\gamma = 4$ meV and $\omega_p = 4.7$ meV, which corresponds to a carrier mobility ($\mu$) in the range $\mu \sim 1100 - 2300$ cm$^2$/(Vs). By tracking the time evolution of the conductivity, we find that the scattering rate remains constant (Supplementary Fig. 8). This implies that the carriers are cold and lie around the band edges.

**Role of spin–orbit-entangled excitons in the photoinduced state.** In the following, we show that the excitons are solely responsible for creating this anomalous antiferromagnetic conducting state. First, we examine how the differential signal varies with the absorbed pump laser fluence. In order to maintain a high accuracy, we focus on the spectrally-integrated response by fixing the THz time at the peak of the THz waveform and scanning only the pump delay stage. The pump-induced change in the THz electric field ($\Delta E$) shows the magnon oscillation on top of an exponential relaxation due to the presence of the itinerant carriers

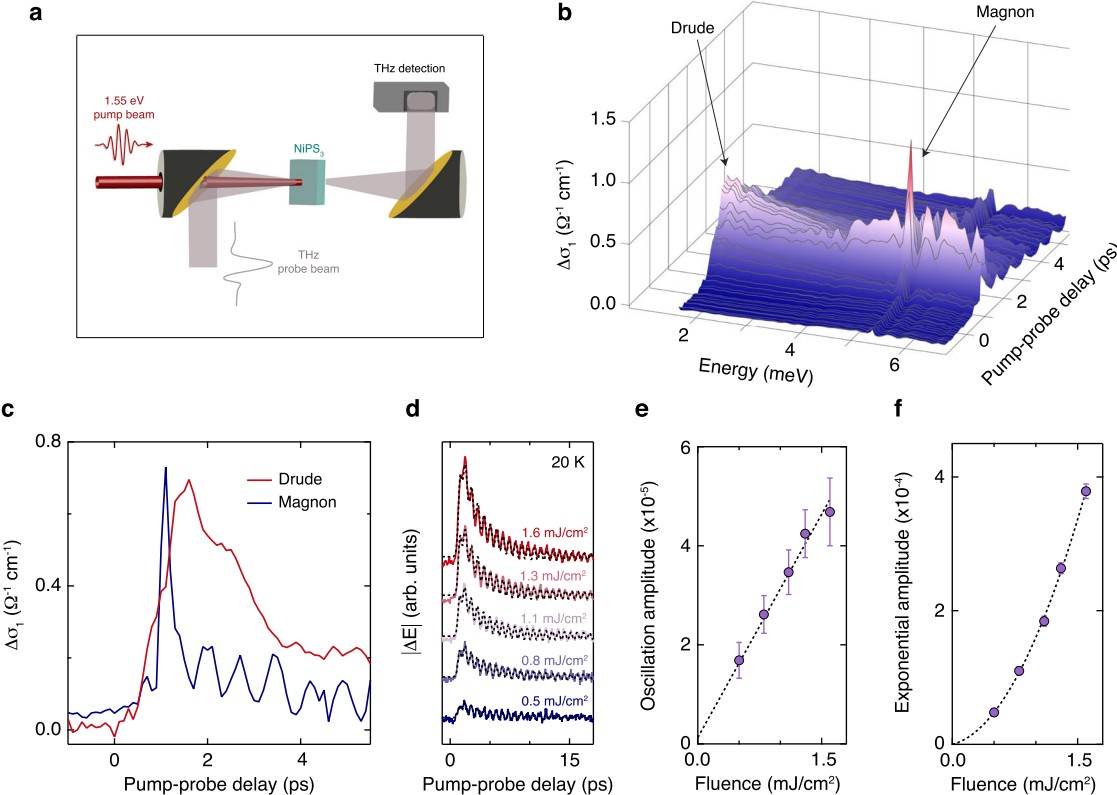

**Fig. 2 Observation of an itinerant conductivity and a coherent magnon upon photoexcitation of the spin–orbit-entangled excitons. a** Schematic of the experimental setup for the THz transmission measurements. **b** Spectro-temporal evolution of the pump-induced change in the real part of the optical conductivity ($\Delta\sigma_1$). The temperature is 20 K and the absorbed pump fluence is 1.3 mJ/cm$^2$. The two features present in the data are a Drude-like response at low energies and the first derivative of a Lorentzian lineshape around the magnon energy. **c** Temporal evolution of $\Delta\sigma_1$ showing coherent magnon oscillations that begin during the rise of the Drude. **d** Temporal evolution of the pump-induced change in the THz electric field ($\Delta E$) of the spectrally-integrated measurement as a function of absorbed pump fluence at 20 K. The black dashed lines are fits to the sum of a damped oscillation and an exponential background. The traces are offset vertically for clarity. **e** The amplitude of the oscillation as a function of fluence extracted from the fits. The oscillation amplitude varies linearly with fluence, indicating that it is proportional to the density of photogenerated excitons. **f** The amplitude of the exponential as a function of fluence. The quadratic behavior is due to exciton dissociation. The error bars in (**e**) and (**f**) represent the 95% confidence interval for the corresponding fit parameters.

(Fig. 2d). Fitting each trace to the sum of a damped oscillation and an exponential background (black dashed lines) reveals that the amplitude of the oscillation varies linearly with the absorbed fluence (Fig. 2e) while the amplitude of the Drude scales quadratically (Fig. 2f). The linear scaling of the magnon oscillation amplitude indicates that it is proportional to the density of photoexcited excitons. On the other hand, the quadratic dependence of the Drude response is rather anomalous. Two different phenomena can generate itinerant carriers with such a fluence dependence[10]: direct two-photon absorption[11] or exciton dissociation[12–15]. Given that our pump photon energy lies in the vicinity of exciton transitions, two-photon absorption through a virtual state is very unlikely. This is confirmed by the linear scaling of the magnon amplitude on the absorbed laser fluence. Further evidence against two-photon absorption is found by extending our experiments to $FePS_3$ and $MnPS_3$, two other compounds closely related to $NiPS_3$. Pumping below the optical gap but far from any exciton transition yields no photoinduced THz signal (see Supplementary Note 8). Thus, exciton dissociation is the source of the quadratic Drude response in $NiPS_3$. Possible microscopic mechanisms for the dissociation process are annihilation via exciton-exciton interactions[12,14,15] and pump-induced exciton photoionization[13].

Second, we elucidate the ultrafast dynamics of the coherent magnon and establish its connection to the photogenerated spin–orbit-entangled excitons. We detect the THz radiation emitted by the sample in the absence of an incident THz probe (Fig. 3a). This signal yields direct signatures of coherent magnons when dipoles in the sample oscillate perpendicular to the direction of light propagation[16], as is the case here, and is devoid of any electronic response. Figure 3b shows the emitted THz electric field ($E$) as a function of time after the pump pulse arrival at various temperatures. The signal consists of a single-frequency oscillation that persists for a long time at low temperature and becomes increasingly more damped as the magnetic transition is approached. A Fourier transform analysis reveals that the oscillation energy softens towards $T_N$ (Fig. 3c), confirming the magnetic origin of this coherent collective mode. Comparing this scaling of the magnon energy to that of the equilibrium case at all temperatures reveals that the energy in the driven system is redshifted (Fig. 3d), which corroborates our finding in Fig. 2b. Furthermore, the phase of the oscillation is independent of the linear or circular polarization state of the pump (see Supplementary Fig. 10 and Supplementary Note 5A), ruling out a conventional mechanism for coherent magnon generation based on impulsive stimulated Raman scattering[17]. Fixing the temperature, we examine the dependence of the coherent magnon energy on the absorbed pump fluence. The results are shown in Figs. 3e and f for 20 K and 110 K, respectively. We observe that the magnon energy decreases linearly as a function of fluence, with

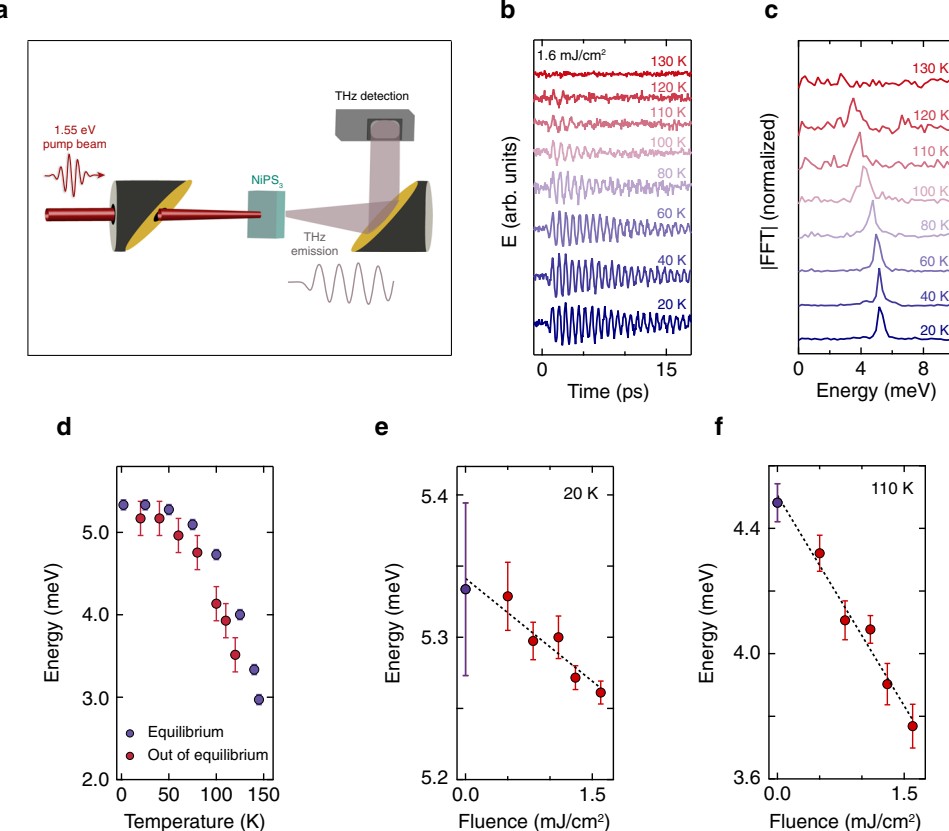

**Fig. 3 Evidence for the involvement of the spin–orbit-entangled excitons in the coherent magnon generation. a** Schematic of the experimental setup for the THz emission measurements. **b** Temporal evolution of the emitted THz electric field ($E$) at various temperatures showing coherent magnon oscillations. The absorbed pump fluence is 1.6 mJ/cm². **c** Fourier transform of the traces in (**b**). A softening of the energy is observed as the temperature approaches $T_N$. The traces in (**b**) and (**c**) are offset vertically for clarity. **d** Comparison of the magnon energies from the measurements performed in equilibrium (Fig. 1c) and out of equilibrium (**c**), revealing that the energy of the driven system is redshifted. The error bars for the violet points represent the energy resolution of the time-domain THz experiment, and those for the red points indicate the resolution of the Fourier transform of the THz emission traces. **e, f** Absorbed pump fluence dependence of the energy of the magnon oscillations at 20 K (**e**) and 110 K (**f**) obtained from fits to the THz emission data (red points). The violet points correspond to the magnon energies in equilibrium at these two temperatures. The energy decreases linearly in both cases. The error bars in (**e**) and (**f**) denote the 95% confidence interval of the fits.

the data points at 110 K exhibiting a steeper slope due to the more fragile nature of the antiferromagnetic order closer to $T_N$. This linear dependence on laser fluence suggests that the magnon energy is also proportional to the density of photogenerated excitons. Since the redshift is non-thermal in nature (see Supplementary Fig. 11 and Supplementary Note 5B), the excitons are responsible for the coherent magnon dynamics. We remark that the lack of broadening of the magnon lineshape (Fig. 2b) as well as the scaling of the magnon amplitude with absorbed fluence (Fig. 2e) rules out the possibility that the signal originates from the nucleation of conducting patches within the antiferromagnetic insulating background[18,19] (see Supplementary Note 7 for details regarding the scenario of phase separation).

## Discussion

Combining all the observables monitored by our experiment, we can now describe the ultrafast dynamics following photoexcitation of the spin–orbit-entangled excitons in $NiPS_3$ (Fig. 4). Upon light absorption, excitons are created (the calculated exciton wavefunction is depicted in purple and green) and their strong coupling to the antiferromagnetic background is key to launching long-wavelength coherent magnon oscillations. Simultaneously, the excitons dissociate into mobile carriers with small excess energies. This results in a homogeneous itinerant conductivity while the underlying long-range antiferromagnetic order is preserved. Such a direct observation of both mobile carriers and long-range spin correlations is uniquely possible in our time-resolved THz experiment due to the concomitant mapping of the Drude response and the zone-center magnon lineshape with high energy resolution.

The emergence of a photoinduced conducting antiferromagnetic state is highly unusual and has not previously been reported in an undoped Mott insulator, where such a phase does not exist at any temperature in equilibrium. Our photoexcitation scheme of pumping below-gap spin–orbit-entangled excitons also profoundly differs from one that relies on photodoping hot particle-hole pairs across the Mott gap. In the latter case, the injected quasiparticles release their excess energy through the emission of hot optical phonons and high-frequency magnons, which leads to the melting of long-range antiferromagnetic correlations while transient metallicity is established[20–22]. This

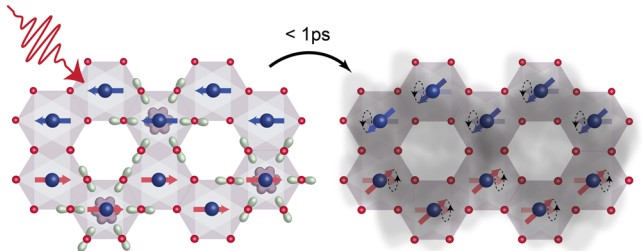

**Fig. 4 Cartoon of the photoinduced dynamics.** A near-infrared pump pulse excites the spin--orbit-entangled excitons in $NiPS_3$ (left panel). The purple shape around the Ni atoms (blue spheres) represents the region where the electron density is confined, whereas the light green structures around the S atoms (red spheres) define where the hole density is distributed. These wavefunctions are computed considering the difference in electron density between the ground state with $^3T_{2g}$ symmetry and the excited state with $^1A_{1g}$ symmetry using configuration interaction theory. Through their coupling to the underlying antiferromagnetic order, the excitons launch a coherent spin precession (denoted by the circular motion of the red and blue arrows, right panel). Within 1 ps, the excitons also dissociate into itinerant carriers yielding a conducting state (gray shading) on top of the persisting long-range antiferromagnetic order.

quench of the long-range magnetic order proceeds through the destruction of the local moments, the reduction of the effective exchange interaction, or the thermal transfer of energy from the electronic carriers and phonons to the spins[23]. In contrast, the spin–orbit-entangled excitons in $NiPS_3$ display an extremely narrow linewidth[5], implying that they retain a high degree of coherence for at least several picoseconds. Also, as mentioned above, our data show that the free carriers generated by exciton dissociation are cold before becoming trapped at defect centers. All of these phenomena prevent the transfer of energy from hot optical phonons to the magnetic order through phonon-spin relaxation channels and thus protect long-range antiferromagnetism (see Supplementary Notes 4 and 6 for further discussion). Our results demonstrate the need for the future development of advanced theoretical methods to investigate the detailed interactions of a non-equilibrium population of excitons in a correlated environment. Currently, the study of excitonic interactions in the dynamics of Mott insulators is still in its infancy.

This mechanism of pumping spin–orbit-entangled excitons in $NiPS_3$ and investigating the resulting magnetic and electronic responses offers a powerful protocol for studying many-body exciton physics. Since excitons living in a correlated environment can also experience strong coupling to degrees of freedom other than spin, this mechanism can be extended to realize other exotic types of collective coherent control or novel phases of matter. While in $NiPS_3$ the photoexcitation of excitons coupled to the magnetic order triggers a coherent magnon, in correlated materials with other types of exciton couplings it will lead to the tailored modulation of, for example, orbital order where promising candidates are the Mott-Hubbard excitons of titanates[24] and vanadates[25]. Note: During the review process, we became aware of another ultrafast study on $NiPS_3$[26].

## Methods

**Single crystal growth and characterization.** High-quality single crystals of $NiPS_3$, $FePS_3$, and $MnPS_3$ were grown by chemical vapor transport in an argon atmosphere using a stoichiometric ratio of elemental powders along with an additional 5% of sulfur[27]. The $NiPS_3$ single crystal used in the THz measurements had a thickness of 1.2 mm and lateral dimensions of ~5 by 5 mm. To characterize our $NiPS_3$ crystal, we measured the in-plane magnetic susceptibility as a function of temperature (Supplementary Fig. 1). The derivative of the susceptibility reveals that the magnetic transition temperature for our sample is $T_N \sim$ 157 K, which is very similar to that of previous reports[7,28]. We also performed heat capacity measurements as a function of temperature (Supplementary Fig. 2). Likewise, the heat capacity exhibits an anomaly at $T_N$.

**Time-domain and ultrafast THz spectroscopy.** We used a Ti:Sapphire amplified laser system that emits 100 fs pulses at a photon energy of 1.55 eV and a repetition rate of 5 kHz to generate THz pulses via optical rectification in a ZnTe crystal. For the time-domain THz spectroscopy experiment, we used electro-optic sampling in a second ZnTe crystal to detect the THz signal transmitted through the sample by overlapping the THz field with a gate pulse at 1.55 eV. We determined the frequency-dependent complex transmission coefficient by comparing the THz electric field through our $NiPS_3$ crystal to that through a reference aperture of the same size (the apertures used for the sample and reference were each 2 mm in diameter). From the measured complex transmission coefficient, we numerically extracted the real and imaginary part of the refractive index[29] from which we determined the absorption coefficient of $NiPS_3$ as a function of frequency in the THz range. Further details can be found in Supplementary Note 2.

For the out-of-equilibrium experiments (THz transmission and THz emission), a fraction of the laser output was used as a pump beam at 1.55 eV. In the THz emission measurements, the THz field spontaneously radiated by the sample upon photoexcitation was subsequently detected using the scheme described above. In contrast, in the THz transmission experiment, a delayed, weak THz probe was transmitted through the photoexcited sample and then detected. The spot size of the pump beam was 4 mm in diameter at the sample position to ensure a uniform excitation of the area probed by the THz pulse (~1 mm in diameter). The time delay between pump and probe (pump delay stage) and the time delay between the THz probe and the gate pulse (gate delay stage) could be varied independently. To measure the spectrally-integrated response, the THz time was fixed at the peak of the THz waveform and the pump delay stage was scanned. The spectrally-resolved

measurements were obtained by synchronously scanning the pump and gate delay stages such that the pump-probe delay time remained fixed, in order to avoid any frequency-dependent artifacts in the non-equilibrium spectra[30]. An optical chopper was placed in the pump beam path to allow for the measurement of the THz electric field in the presence/absence of the pump pulse. The data analysis procedure for extracting the pump-induced change in the optical properties of $NiPS_3$ is discussed in Supplementary Note 2.

**Theoretical calculations**. To characterize the real-space spin precession involved in the observed magnon mode at the Brillouin zone center, we considered an effective spin model of spin-1 Ni sites with no orbital degeneracy. Assuming the interlayer coupling is small, for each layer the effective spin Hamiltonian consists of *XXZ* terms up to third-nearest neighbors and single-ion anisotropy along the *a*- and the *c*-axis. We obtained the magnon dispersion relation by applying the Holstein-Primakoff transformation and block-diagonalizing the resulting Hamiltonian. The details of the computation are presented in Supplementary Note 9.

To determine the exciton wavefunction, we performed a configuration interaction calculation of an $NiS_6$ cluster[5]. The calculation included all possible states with $d^8$, $d^9\underline{L}^1$, and $d^{10}\underline{L}^2$ configurations, where $\underline{L}$ refers to the ligand S $3p$-hole orbitals. We took into account the cubic crystal field and the on-site Coulomb interaction for Ni $3d$ orbitals, which are parameterized with Slater-Condon parameters ($F^0$, $F^2$, $F^4$). We assumed that all the ligand $3p$ orbitals have the same energy levels characterized by the charge-transfer energy, and the hopping integrals between the Ni $3d$ and S $3p$ orbitals were determined with two parameters $V_{pd\sigma}$ and $V_{pd\pi}$ using the Slater–Koster theory. The spatial distribution of the spin–orbit-entangled excitons was computed from the difference in electron density between the ground state with $^3T_{2g}$ symmetry and the excited state with $^1A_{1g}$ symmetry.

## Data availability

The data that support the findings of this study are available from the corresponding authors upon reasonable request.

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

## Acknowledgements

We acknowledge useful discussions with Jonathan Pelliciari and Ki Hoon Lee. Work at MIT was supported by the US Department of Energy, BES DMSE (data taking and analysis), and by the Gordon and Betty Moore Foundation's EPiQS Initiative grant GBMF9459 (instrumentation). C.A.B. and E.B. acknowledge additional support from the National Science Foundation Graduate Research Fellowship under Grant No. 1745302 and the Swiss National Science Foundation under fellowships P2ELP2-172290 and P400P2-183842, respectively. D.M. and T.S. are supported by a US Department of Energy grant DE-SC0008739, and T.S., in part, by a Simons Investigator award from the Simons Foundation. H.C.P. is supported by a Pappalardo Fellowship at MIT and a Croucher Foundation Fellowship. Work at IBS-CCES was supported by the Institute for Basic Science (IBS) in Korea (Grant No. IBS-R009-G1) and work at CQM was supported by the Leading Researcher Program of the National Research Foundation of Korea (Grant No. 2020R1A3B2079375). J.H.K. acknowledges support from the National Research Foundation of Korea (NRF) grants funded by the Ministry of Science, ICT, and Future Planning (MSIP) of Korea (NRL Program No. NRF-2019R1I1A2A01062306, SRC Program No. NRF-2017R1A5A1014862).

## Author contributions

C.A.B., E.B., I.O.O., and C.J.A. performed the THz experiments. C.A.B. and E.B. analyzed the experimental data. S.S., I.H., and J.-G.P. grew the single crystals. J.K and J.H.K. carried out the optical transmission measurements. D.M., H.C.P., and T.S. performed the magnon calculations, and B.H.K. performed the exciton calculations. C.A.B., E.B., J.-G.P., and N.G. wrote the manuscript with crucial input from all other authors. This project was supervised by N.G.

## Competing interests

The authors declare no competing interests.
