## [Peer Review File · Nature Communications]

Reviewers' Comments:

Reviewer #1:

Remarks to the Author:

The manuscript by Belvin and collaborators presents THz measurement of the zone-center magnon and Drude charge response of NiPS₃ after photo-excitation tuned to a recently discovered Zhang-Rice exciton. The work is interesting and there are many nice aspects to it. The main selling point is a transient metallic state that preserves long-range antiferromagnetic correlations and the assertion that this state is highly novel. I agree that with the concept that if one introduces electron-hole pairs into a Mott insulator then they are liable to hop around and efficiently destroy magnetic order. While I see significant potential for the work, I feel that the manuscript comes up a bit short in its description of the state generated. I was expecting some ballpark estimates of how long the magnetic correlation length is in the transient state and similar ballpark estimates of how many excitons and how many itinerant carriers there might be. I have no problem if these estimates are a bit crude, but I find it troublesome that such estimates are omitted entirely. The reason I ask is that if I imagine, for the sake of argument, that only a very small number of itinerant carriers were present, separated by \gg the magnetic correlation length, then the case for impact would be much less clear. Below I ask three questions about this topic and then make six further less important comments.

1. Is it possible to be clearer about the initial number density of excitons (just from the fluence and excited volume)? The manuscript leaves it quite late to mention they have ps lifetime. It's a bit hard for me to understand how the time evolution of the Drude response can be rationalized if excitons are decaying over the ps measurement window.
2. Crudely speaking, the magnetic correlation length should scale like \sim magnon wavelength *magnon energy/ magnon linewidth. Can the manuscript include this (or a more precise) estimation of the magnetic correlation length in the transient state?
3. Some discussion of how many itinerant carriers there might be is needed. With this information (or with clear statements in the manuscript about what is not known) in hand, I will be able to make an impact assessment.

Some smaller things

4. The sentence

"Consequently, the electron spins are also localized and order antiferromagnetically at low temperature due to entropy and energy considerations."
is not really correct. Entropy isn't why antiferromagnetism forms and to say it forms due to energy considerations does not mean much. I am not sure an explicit explanation is needed, but "Consequently, the electron spins are also localized and order antiferromagnetically at low temperature in order to minimize their kinetic energy." would be an appropriate, if very brief, explanation if desired.

5. The way Ref 10 and ref 11 are used on page 6 is, for me, not so appropriate. I think the authors mean to invoke very simple concepts and not really to draw analogies with these other materials. It would be better to include physical arguments.

6. I tend to agree with the sentence

"Notably, these oscillations begin during the rise of the Drude response, indicating that a nonthermal mechanism is responsible for launching the magnon coherently."
But it would be better to state the authors' reasoning. Is it simply timescale?

7. The manuscript is overall well written, but I would ask the authors to reconsider the way they use the supplementary information. The purpose of a couple of sections is not so clear and I feel that "Supplementary Note 5" is more like the type of material that should be in the main manuscript. If the authors think the organization is appropriate, I do not want to insist on changes here, just to raise the question.

8. Some of the theory presented is strange. There are 11 pages of spin-wave theory. I was happy to read it, but I was constantly distracted by wondering what the purpose was. Is this used purely to generate the cartoon of the zero-energy magnon? Is this information that is not available in the literature already?

9. I didn't understand the purpose of the cluster calculation section. It seems rather similar to what is presented in the Kang et al. Nature paper, although I vaguely remember them taking into account crystal field more precisely. In any case, what's presented isn't enough to convey a picture of what's going on. At an absolute minimum the reader would need to know the hole character and spin state, these aren't described and depend on unstated parameters related to the Columb and charge transfer energy. I would like to raise the question of whether it would be best removed.

Reviewer #2:

Remarks to the Author:

The authors have presented quite comprehensive study of the exciton-driven antiferromagnetic metal in NiPS₃. I believe that the authors have reached an important conclusion of the coexistence of itinerant carriers produced by exciton dissociation and the long-wavelength antiferromagnetic magnon that coherently precesses in time. Given the importance and novelty, I am in strong support of accepting this manuscript before the following comments are addressed:

1. Is there any temperature limitation of the main observation? Fig. 2b was measured at T=20K? Has the author performed more measurements at different temperature?
2. I am curious about the XXZ model used for theoretical calculation. As the authors wrote in the main text, the magnon at Γ point is observed ~ 5.3 meV at equilibrium state. However, they gave a magnon gap $\sim 6-7$ meV in the Supp. I am confused about all these numbers. The authors need to explain more.

Reviewer #3:

Remarks to the Author:

The authors report on an ultrafast study of the van der Waals correlated insulator NiPS₃, which was recently shown to exhibit spin-orbit entangled excitons as unusual bound states in a correlated many-body system. THz detection is used to reveal both the low-energy magnon resonances of NiPS₃ in equilibrium and the time-dependent dynamics of its THz optical conductivity after excitation near the exciton line. Coherent magnon oscillations are observed, whose excitation mechanism is assigned to exciton-magnon coupling via the scaling with pump fluence. Moreover, the authors observe a transient metallic phase, which coexists with the AF order, representing a non-equilibrium phase outside the thermally accessible phase diagram.

In my view, the work provides an exciting new perspective on the dynamic coupling of excitonic and spin-orbit many-body states in this correlated system. It should be of interest to an extended community and will also motivate new theoretical exploration of entangled orders involving excitonic degrees of freedom in strongly correlated materials. The latter is currently not well understood and will benefit from new efforts both experimentally (as in the current study) and theoretically. Overall, based on the level of novelty and impact on the field, driving forward the study of many-body

excitonic phases and their dynamics based on this intriguing system, I recommend publication in Nature Communications.

Reviewer #1 (Remarks to the Author):

The manuscript by Belvin and collaborators presents THz measurement of the zone-center magnon and Drude charge response of NiPS₃ after photo-excitation tuned to a recently discovered Zhang-Rice exciton. The work is interesting and there are many nice aspects to it. The main selling point is a transient metallic state that preserves long-range antiferromagnetic correlations and the assertion that this state is highly novel. I agree that with the concept that if one introduces electron-hole pairs into a Mott insulator then they are liable to hop around and efficiently destroy magnetic order.

While I see significant potential for the work, I feel that the manuscript comes up a bit short in its description of the state generated. I was expecting some ballpark estimates of how long the magnetic correlation length is in the transient state and similar ballpark estimates of how many excitons and how many itinerant carriers there might be. I have no problem if these estimates are a bit crude, but I find it troublesome that such estimates are omitted entirely. The reason I ask is that if I imagine, for the sake of argument, that only a very small number of itinerant carriers were present, separated by \gg the magnetic correlation length, then the case for impact would be much less clear. Below I ask three questions about this topic and then make six further less important comments.

We thank the Referee for their thorough review of our paper and for appreciating the significance of our findings. Below we address each of the Referee's concerns, and we hope the Referee will now find our manuscript suitable for publication in Nature Communications.

1. Is it possible to be clearer about the initial number density of excitons (just from the fluence and excited volume)? The manuscript leaves it quite late to mention they have ps lifetime. It's a bit hard for me to understand how the time evolution of the Drude response can be rationalized if excitons are decaying over the ps measurement window.

We thank the Referee for pointing this out as we did not mention the density of photogenerated excitons in the previous version of our manuscript. Using the absorbed pump fluence and photoexcited volume, we calculate that the initial density of excitons is on the order of 10^{19} cm^{-3} . A fraction (see our answer to Question 3 below) of these excitons subsequently undergo dissociation, giving rise to itinerant carriers that produce the Drude response. Therefore, in the photoinduced state there is a coexistence of excitons (that have not dissociated) and mobile carriers. These two populations are likely to follow independent dynamics and thus there is no contradiction between the time evolution of the Drude response and the exciton coherence lifetime. Moreover, we note that the exciton coherence lifetime is at least 10 ps (from S. Kang et al. Nature 583, 785 (2020)). This lower bound is set by the linewidth of the exciton resonance in the equilibrium optical absorption spectrum (0.4 meV), which is limited by the experimental resolution and also accounts for possible inhomogeneous broadening. Thanks to this valuable comment, we clarified these aspects in the revised version of our paper in Supplementary Note 3 on page 10.

2. Crudely speaking, the magnetic correlation length should scale like \sim magnon wavelength *magnon energy/magnon linewidth. Can the manuscript include this (or a more precise) estimation of the magnetic correlation length in the transient state?

The magnetic correlation length that is relevant for our experiment is the one associated with *static* magnetic order and therefore is the one extracted from the linewidth of the magnetic Bragg peak in diffraction experiments. For single crystals of NiPS₃, there are currently no elastic neutron/x-ray scattering data from which this static correlation length can be estimated. However, in our experiment, we note that the magnon mode we observe is the lowest-energy (“pseudo-Goldstone”) mode at $q = 0$. As such, it is a robust fingerprint of the long-range antiferromagnetic order. In the transient state, there is no broadening of this magnon mode, thus confirming that the static long-range magnetic order is preserved and is not modified significantly by the presence of the itinerant carriers. In practice, the magnetic correlation length is limited by the size of the magnetic domains in the crystal. In MPS₃ materials (where M is a transition metal), these domains have a size on the order of a few μm (H. Chu et al., Phys. Rev. Lett. 124, 027601 (2020)).

Moreover, we have demonstrated that our photoexcited state is homogeneous (see Supplementary Note 7) so the itinerant carriers form a homogeneous conducting phase within the pump excitation spot size. This allows us to estimate that the itinerant charge carriers are separated by $\sim 120\text{-}150$ nm (based on the value of the carrier density – see our answer to Question 3 below). Therefore, we can conclude that the carriers are separated by a distance that is much smaller than the magnetic correlation length. We have updated Supplementary Note 7 in order to account for this important comment by the Referee.

3. Some discussion of how many itinerant carriers there might be is needed. With this information (or with clear statements in the manuscript about what is not known) in hand, I will be able to make an impact assessment.

Using the extracted plasma frequency from fits to the Drude response (see Supplementary Note 4) and the range of effective masses (determined from electronic structure calculations in C. Lane and J.-X. Zhu, Phys. Rev. B 102, 075124 (2020)), we estimate that the density of itinerant carriers is in the range of 3×10^{14} to 6×10^{14} cm^{-3} . Since the conducting phase is homogeneous (see Supplementary Note 7), a uniform distribution of carriers with these densities implies that the carriers are separated by 120-150 nm. Moreover, the itinerant carriers have a very high mobility of 1100-2300 cm^2/Vs (see Supplementary Note 4), which defines how conducting the state is. We remark that the density can be further increased (in a quadratic manner) by increasing the pump fluence. In our state-of-the-art setup in which we probe the THz conductivity, this is challenging because the pump spot size must be larger than the THz probe size to ensure a uniform illumination. We have added this information to Supplementary Note 4 on page 13.

Some smaller things

4. The sentence “*Consequently, the electron spins are also localized and order antiferromagnetically at low temperature due to entropy and energy considerations.*” is not really correct. Entropy isn’t why antiferromagnetism forms and to say it forms due to energy considerations does not mean much. I am not sure an explicit explanation is needed, but “*Consequently, the electron spins are also localized and order antiferromagnetically at low temperature in order to minimize their kinetic energy.*” would be an appropriate, if very brief, explanation if desired.

We agree with the Referee that the preferential formation of antiferromagnetism over ferromagnetism is due to the minimization of kinetic energy. In our original sentence we wanted to emphasize that the ordering occurs in the first place because, in the absence of order, the state of the system would be highly degenerate, contradicting the Nernst theorem (see Chapter 1 of D. I. Khomskii *Transition Metal Compounds*). We realize that by compressing these ideas into a single sentence our meaning was not clear. To address the Referee's comment, we added one more sentence explaining the role of the kinetic energy (page 3 of the main text).

5. The way Ref 10 and ref 11 are used on page 6 is, for me, not so appropriate. I think the authors mean to invoke very simple concepts and not really to draw analogies with these other materials. It would be better to include physical arguments.

The Referee is right in that the two sentences should refer to general physical arguments rather than specific comparisons with other materials. We removed the references from page 6 and described only the physical concepts.

6. I tend to agree with the sentence “*Notably, these oscillations begin during the rise of the Drude response, indicating that a nonthermal mechanism is responsible for launching the magnon coherently.*” But it would be better to state the authors' reasoning. Is it simply timescale?

The Referee is correct that in the quoted sentence we are referring primarily to the timescale. If the magnon were launched thermally, its oscillations would begin later, only after thermalization with the lattice (see, for example, Fig. 2 of A. V. Kimel et al. *Nature* 429, 850 (2004)). To clarify our intended meaning, we have rephrased this sentence on page 6.

We emphasize that, in addition to the timescale, further evidence supporting the non-thermal scenario is given by the pump fluence dependence of the magnon energy, which we discuss later in the paper on page 9 and also in more detail in Supplementary Note 5B. The decrease in magnon energy with fluence is significantly larger than the change that would occur thermally from the increase in lattice temperature due to the heat deposited by the pump pulse.

7. The manuscript is overall well written, but I would ask the authors to reconsider the way they use the supplementary information. The purpose of a couple of sections is not so clear and I feel that “Supplementary Note 5” is more like the type of material that should be in the main manuscript. If the authors think the organization is appropriate, I do not want to insist on changes here, just to raise the question.

We agree with the Referee that the content of Supplementary Note 5 (now Supplementary Note 6 in the revised version) is important for understanding the significance of our paper, particularly for researchers who are more familiar with band semiconductors than strongly correlated systems. We have included it in the Supplementary Information in order to not break up the flow of the main and to keep the discussion section concise. We have added a statement to our Discussion referring the reader to this Supplementary Note for additional details.

To improve the organization of the Supplementary Information, we have restructured some of the sections and added a table of contents at the beginning along with brief descriptions of each

section. We hope that this makes it easier for the reader to locate information and to understand the purpose of each section.

8. Some of the theory presented is strange. There are 11 pages of spin-wave theory. I was happy to read it, but I was constantly distracted by wondering what the purpose was. Is this used purely to generate the cartoon of the zero-energy magnon? Is this information that is not available in the literature already?

The Referee is correct that the main purpose of the spin wave theory is to calculate the energy of the magnon gap at the Γ point, which we find agrees well with the experimentally observed value, and to determine the real-space precession of the spins corresponding to this magnon mode. While a preliminary spin wave calculation for NiPS₃ is given in the literature (see K. Kim et al. Nat. Commun. 10, 345 (2019)), a thorough discussion of the symmetries of the zigzag magnetic order has not been provided in other studies. Therefore, we thought it would be valuable to include this in our Supplementary Information. However, we agree with the Referee that the reader can be distracted from the main purpose of the section. For this reason, we added a short description in the table of contents under Supplementary Note 9 to clarify the ultimate goal of our analysis.

9. I didn't understand the purpose of the cluster calculation section. It seems rather similar to what is presented in the Kang et al. Nature paper, although I vaguely remember them taking into account crystal field more precisely. In any case, what's presented isn't enough to convey a picture of what's going on. At an absolute minimum the reader would need to know the hole character and spin state, these aren't described and depend on unstated parameters related to the Coulomb and charge transfer energy. I would like to raise the question of whether it would be best removed.

The cluster calculation is indeed the same as that presented in S. Kang et al. Nature 583, 785 (2020), the work of several of our coauthors. In the current paper, we only gave a brief description in the Methods section of the essential parameters and left the details to that previous work. The main purpose of including it was to explain how the exciton wavefunctions depicted in Fig. 4 are calculated. Following the Referee's suggestion, we removed the details of the cluster calculation section and simply refer the reader to S. Kang et al. Nature 583, 785 (2020).

Reviewer #2 (Remarks to the Author):

The authors have presented quite comprehensive study of the exciton-driven antiferromagnetic metal in NiPS₃. I believe that the authors have reached an important conclusion of the coexistence of itinerant carriers produced by exciton dissociation and the long-wavelength antiferromagnetic magnon that coherently precesses in time. Given the importance and novelty, I am in strong support of accepting this manuscript before the following comments are addressed:

We thank the Referee for acknowledging the significance of our work.

1. Is there any temperature limitation of the main observation? Fig. 2b was measured at T=20K? Has the author performed more measurements at different temperature?

Yes, there is a temperature limitation. It is given by the Néel temperature ($T_N = 157$ K), below which the system forms long-range antiferromagnetic order. In Fig. 2b, the data are presented at low temperature (20 K) because that is where the spin ordering is most robust and the signal-to-noise ratio of our measurement is the highest. However, we have performed systematic measurements at different temperatures, observing a similar coexistence of the Drude response with the long-wavelength antiferromagnetic magnon below T_N (see Supplementary Fig. S9b for the spectrally-integrated pump-probe response and Fig. 3b for the behavior of the magnon alone). The amplitude of the Drude signal is related to the temperature dependence of the excitonic absorption (see Fig. 2d of S. Kang et al. Nature 583, 785 (2020)), as the spin-orbit-entangled excitons (that subsequently dissociate) are coupled to the long-range antiferromagnetic order and emerge with finite oscillator strength only below T_N . Thanks to this comment by the Referee, we clarified this aspect in Supplementary Note 4 on page 19 of our revised version.

2. I am curious about the XXZ model used for theoretical calculation. As the authors wrote in the main text, the magnon at Γ point is observed ~ 5.3 meV at equilibrium state. However, they gave a magnon gap $\sim 6-7$ meV in the Supp. I am confused about all these numbers. The authors need to explain more.

The discrepancy between the calculated and experimental values of the magnon gap lies in the parameters of the Hamiltonian. In our calculations, these parameters (i.e. exchange couplings and magnetocrystalline anisotropies) were taken from a previous neutron scattering study (D. Lançon et al. Phys. Rev. B 98, 134414 (2018)), which offered a preliminary overview of the magnetic excitations in NiPS₃ but had a limited energy coverage and finite energy resolution. Future measurements with an increased energy window and resolution would refine the parameters of the Hamiltonian, leading to a more accurate estimate of the magnon gap we observe at Γ via THz spectroscopy. We remark that our THz technique has a very high energy resolution ($\Delta E \sim 60$ μ eV), but it is limited to first order only to $q = 0$ excitations. Therefore, it alone cannot be used to estimate the complete parameter set of the XXZ Hamiltonian. To account for this interesting comment by the Referee, we added some sentences in Supplementary Note 9D on page 37.

Reviewer #3 (Remarks to the Author):

The authors report on an ultrafast study of the van der Waals correlated insulator NiPS₃, which was recently shown to exhibit spin-orbit entangled excitons as unusual bound states in a correlated many-body system. THz detection is used to reveal both the low-energy magnon resonances of NiPS₃ in equilibrium and the time-dependent dynamics of its THz optical conductivity after excitation near the exciton line. Coherent magnon oscillations are observed, whose excitation mechanism is assigned to exciton-magnon coupling via the scaling with pump fluence. Moreover, the authors observe a transient metallic phase, which coexists with the AF order, representing a non-equilibrium phase outside the thermally accessible phase diagram.

In my view, the work provides an exciting new perspective on the dynamic coupling of excitonic and spin-orbit many-body states in this correlated system. It should be of interest to an extended community and will also motivate new theoretical exploration of entangled orders involving excitonic degrees of freedom in strongly correlated materials. The latter is currently not well understood and will benefit from new efforts both experimentally (as in the current study) and theoretically. Overall, based on the level of novelty and impact on the field, driving forward the study of many-body excitonic phases and their dynamics based on this intriguing system, I recommend publication in Nature Communications.

We thank the Referee for appreciating the importance of our findings and their impact on a broad community of experimentalists and theorists.

Reviewers' Comments:

Reviewer #1:

Remarks to the Author:

I thank the authors for the revisions and consider that the work is suitable for Nat. Comm.

I would like to ask the authors to consider the following (optional) change

"No sizeable broadening of the lineshape occurs, meaning that the long-range antiferromagnetic order is preserved and there is no coupling between the itinerant electrons and the localized spins."

->

"No sizeable broadening of the lineshape occurs, meaning that the long-range antiferromagnetic order is preserved."

there are not so many itinerant carriers, so whether or not there is coupling is not really tested.

Reviewer #2:

Remarks to the Author:

I want to thank the authors for the detailed reply and comments. My comments and suggestions are addressed and, where suitable, incorporated into the manuscript.

In conclusion, I recommend the manuscript for publication in Nature communications.

Reviewer #1 (Remarks to the Author):

I thank the authors for the revisions and consider that the work is suitable for Nat. Comm.

We thank the Reviewer for appreciating our revisions and for finding our work suitable for publication in Nature Communications.

I would like to ask the authors to consider the following (optional) change

"No sizeable broadening of the lineshape occurs, meaning that the long-range antiferromagnetic order is preserved and there is no coupling between the itinerant electrons and the localized spins."

->

"No sizeable broadening of the lineshape occurs, meaning that the long-range antiferromagnetic order is preserved."

there are not so many itinerant carriers, so whether or not there is coupling is not really tested.

We agree with the Referee and have made this change in our manuscript.

Reviewer #2 (Remarks to the Author):

I want to thank the authors for the detailed reply and comments. My comments and suggestions are addressed and, where suitable, incorporated into the manuscript.

In conclusion, I recommend the manuscript for publication in Nature communications.

We thank the Reviewer for appreciating our revisions and comments and for recommending the publication of our work in Nature Communications.